# Resveratrol as an Inhibitor of the NorA Efflux Pump and Resistance Modulator in *Staphylococcus aureus*

**DOI:** 10.3390/antibiotics12071168

**Published:** 2023-07-10

**Authors:** Madalena Santos, Raquel Santos, Pedro Soeiro, Samuel Silvestre, Susana Ferreira

**Affiliations:** CICS-UBI—Health Sciences Research Centre, University of Beira Interior, 6200-506 Covilhã, Portugal; madalena.santos@ubi.pt (M.S.); raquel.neves.santos@ubi.pt (R.S.); pedro.soeiro@ubi.pt (P.S.); sms@ubi.pt (S.S.)

**Keywords:** *Staphylococcus aureus*, resistance, NorA, resveratrol, efflux inhibition

## Abstract

*Staphylococcus aureus* can exhibit resistance to various antibiotics. Among its resistance mechanisms, the active efflux of antibiotics can be seen as relevant. This study aimed to evaluate the ability of resveratrol to modulate norfloxacin resistance in *S. aureus*. The antimicrobial activity of resveratrol was assessed using the broth microdilution method to determine the minimum inhibitory concentration (MIC). Then, the modulatory effect of resveratrol was evaluated using the MIC determination for the antibiotic or ethidium bromide in the presence and absence of resveratrol at a sub-MIC level. The MIC of norfloxacin against *S. aureus* SA1199B (NorA-overexpressing strain) decreased 16-fold when in the presence of resveratrol, with a similar behavior being observed for ethidium bromide. An evaluation of the ethidium bromide accumulation was also performed, showing that in the presence of resveratrol, the SA1199B strain had augmented fluorescence due to the accumulation of ethidium bromide. Altogether, the results suggested that resveratrol may act by inhibiting NorA. These in vitro data were supported by docking results, with interactions between resveratrol and the NorA efflux pump predicted to be favorable. Our findings demonstrated that resveratrol may modulate norfloxacin resistance through the inhibition of NorA, increasing the effectiveness of this antibiotic against *S. aureus*.

## 1. Introduction

*Staphylococcus aureus* (*S. aureus*) is a Gram-positive bacterium that belongs to the genus *Staphylococcus* and the family *Staphylococcaceae* [1]. It is a frequent colonizer of humans, namely of the nasal passages, nasopharynx, skin and intestines of healthy individuals. Nonetheless, infections caused by this bacterium have been increasing worryingly, with *S. aureus* being one of humans’ most important opportunistic bacterial pathogens and a cause of high morbidity and mortality [2,3,4]. In general, the infectious potential of *S. aureus* may span from mild conditions to severe invasive infections, with its success as a pathogen being associated with the development of three syndromes: superficial skin infections (wounds and abscesses), deep and systemic infections (osteomyelitis, endocarditis, pneumonia and bacteremia) and toxemic syndrome (toxic shock syndrome, food poisoning, scalded skin syndrome) [5]. *S. aureus* has been implicated in a wide array of community and hospital infections, with most of these infections being directly related to its high colonization capacity, the presence of virulence factors and the development of resistance to various antibiotics [2,6,7].

Antibiotic resistance represents a growing threat to worldwide public health [8], presenting a significant impact on the morbidity and mortality rate of infectious diseases [9]. Thus, the development of effective therapies, which may range from new antimicrobial agents to resistance modifiers, becomes essential [10]. 

Among the resistance mechanisms developed by *S. aureus*, the increased expression of efflux pumps is a mechanism to consider, since they actively transport antibiotics to the extracellular medium, reducing their intracellular concentrations [2,3,11]. Efflux pumps are categorized into several families of transporters: the ATP-binding cassette (ABC) family, multidrug and toxic compound extrusion (MATE) family, major facilitator superfamily (MFS), resistance-nodulation-cell-division (RND) family, small multidrug resistance (SMR) family and proteobacterial antimicrobial compound efflux (PACE) family [12]. Within the MFS family, the most-studied efflux pumps in Gram-positive bacteria are NorA (*S. aureus*) and PmrA (*Streptococcus pneumoniae*) [13,14]. The NorA efflux pump, expressed in *S. aureus*, has been associated with resistance to several compounds, with various specific substrates described, including fluoroquinolones, acridine orange, benzalkonium chloride, cetrimide, ciprofloxacin, ethidium bromide, nalidixic acid and tetraphenylphosphonium ions [2,15]. 

In this context, several studies have focused on research into and development of efflux pump inhibitors (EPIs) in order to modulate antibiotic resistance [16,17]. Although several substances have been proposed as potential EPIs, most of them exhibit high toxicity to the host organism or do not meet the necessary requirements to be considered ideal EPI [2]. Therefore, some natural compounds, including plant extracts, essential oils and their isolated compounds, have been reported as excellent alternatives to synthetic products for reversing bacterial resistance mechanisms [2,3,17]. 

Resveratrol (*trans*-3,4,5′-trihydroxystilbene) is a phytoalexin present in several plants and fruits, such as grapes (*Vitis vinifera*), blackberries, blueberries and peanuts, among others [18]. Previous studies indicate that this compound presents significant antioxidant, anti-inflammatory, anticancer and antibacterial activity [18,19,20]. In addition, resveratrol has been shown to act on multiple targets in bacterial cells and to modify the conventional antibiotic activity [18,19,20]. In *S. aureus*, bacteriostatic activity is described for this compound with no detection of resveratrol-induced membrane damage [19,21]. Moreover, resveratrol has increased the susceptibility of several Gram-negative and Gram-positive bacteria, including *S. aureus,* and yeasts to chlorhexidine and benzalkonium chloride, or to streptomycin and ciprofloxacin in multidrug-resistant Gram-negative species, and enhanced the efficacy of aminoglycosides against *S. aureus,* suggesting its potential as an EPI [21,22,23]. In fact, some studies report that this compound may act as an inhibitor of efflux pumps, as suggested for *Escherichia coli*, *Acinetobacter baumannii* and *Arcobacter* species [20,24,25,26]. In Gram-negative bacteria, several studies have suggested that resveratrol EPI activity may be related to the reduction in the expression of efflux systems, including the RND superfamily, PACE family or MFS superfamily efflux pump genes in *E. coli* and *A. baumannii* [24,26,27]. Although some research works have demonstrated the role of resveratrol in the ability to inhibit efflux pumps in some species, the effect of this compound on the NorA efflux pump of *S. aureus* is yet to be evaluated. 

Thus, in this work, we aimed to evaluate the potential modulatory effect of resveratrol on fluoroquinolone resistance in *S. aureus*, focusing on the NorA efflux pump.

## 2. Results

### 2.1. Evaluation of the Modulatory Effect of Resveratrol

The MICs of norfloxacin and ethidium bromide alone or in combination with resveratrol at 0.25× MIC (Table 1) were determined, with a variation in the value that correlates with NorA expression being observed (Figure 1). A positive tolerance-modulating effect was demonstrated when in the presence of resveratrol, significantly decreasing the norfloxacin MIC in the *S. aureus* SA1199, SA1199B and NCTC 8325-4 strains, with the effect being stronger in the NorA-overexpressing *S. aureus* strain SA1199B (16-fold reduction) (Figure 1A). A similar result was found with ethidium bromide, a recognized substrate of efflux pumps (Figure 1B). For the SAK1758 strain, there was no reduction in the MIC of norfloxacin when in the presence of resveratrol; however, when ethidium bromide, a common substrate for the efflux systems, was applied, a higher reduction in its MIC was observed when compared to norfloxacin.

### 2.2. Ethidium Bromide Accumulation

To further validate resveratrol as a potential EPI, we analyzed its ability to enhance the accumulation of ethidium bromide in *S. aureus* SA1199B. As shown in Figure 2A, in the presence of resveratrol, ethidium bromide accumulation was significantly increased, presenting similar values to those obtained for the known EPI control carbonyl cyanide *m*-chlorophenyl hydrazone (CCCP) after 30 min. When comparing the four tested strains, which presented different levels of NorA expression, a significant difference in the fluorescence at 60 min of the accumulation assay was observed for the two wildtype (SA1199 and NCTC 8325-4) and for the NorA-overexpressing (SA1199B) strains (Figure 2B).

### 2.3. Molecular Docking Studies with Resveratrol

For a better understanding of the observed in vitro results, a molecular docking study was performed. Preliminary studies with reserpine (a known inhibitor) were carried out, and all the preferred relative orientations of reserpine obtained within NorA are presented in Figure 3A,B, implying a possible binding site on the efflux pump. Therefore, molecular docking with resveratrol was also performed considering a grid covering this region. In Figure 3C,D, the different resveratrol orientations in the docking with the *S. aureus* NorA system are represented. Overall, the reserpine conformations were mostly stabilized by van der Waals and classic hydrogen bond interactions, and the estimated binding energies (BE) for resveratrol suggest that this compound can effectively bind to NorA (Figure 4 and Table 2). 

### 2.4. Impact on Mutation Frequency and Post-Antibiotic Effect (PAE)

To further assess the potential of resveratrol, we evaluated its impact on resistance emergence using the *S. aureus* SA1199 strain and determined the norfloxacin PAE alone and in combination with resveratrol on *S. aureus* SA1199B. It was found that in the absence of resveratrol, for different norfloxacin concentrations (4×, 8× and 16× MIC), the mutation frequency was between 1.87 × 10^−5^ and 4.03 × 10^−8^. However, in the presence of resveratrol (50 mg/L), there was a decrease for values between 2.16 × 10^−7^ and <5.31 × 10^−10^, respectively (Table 3). Moreover, the PAE values for the 0.25×, 0.5× and 1× MIC of norfloxacin significantly increased when in the presence of resveratrol (25 mg/L) (Table 4). 

## 3. Discussion

*S. aureus* species exhibit a high capacity to adapt to environmental conditions and rapidly develop resistance to antibiotics. One of the mechanisms pertinent to this study and involved in increasing resistance to several antimicrobial agents is the extrusion of antibiotics through efflux pumps [2]. In this sense, compounds with the potential to inhibit efflux pumps may present themselves as a useful approach for reversing or reducing resistance [28]. Therefore, we tried to clarify the modulatory effect of resveratrol on resistance or tolerance to fluoroquinolones in *S. aureus*, focusing on the NorA efflux system. 

Resveratrol was shown to be able to modulate *S. aureus* resistance to norfloxacin, likely through inhibition of the NorA efflux system, since a higher modulatory effect (~16-fold reduction in norfloxacin MIC) occurred for *S. aureus* SA1199B, restoring norfloxacin susceptibility in this strain. In the case of *S. aureus* SAK1758, there was no reduction in the MIC of norfloxacin, which may be associated with the absence of the NorA efflux pump. An ethidium bromide MIC potentiation in the presence of resveratrol was also observed, with higher evidence for the NorA-overexpressing strain. Considering the levels of NorA expression in the strains evaluated, we may suggest that the effect observed is due to efflux pump inhibition. Similar results have been shown for other natural compounds, such as boeravinone B, capsaicin, α-terpinene, eugenol, isoeugenol, sesquiterpenes isolated from *Pilgerodendron uviferum* and several flavonoids including quercetin [2,3,7,28,29,30,31] or even plant extracts or essential oils, such as ethanoic extract from *Bauhinia forficate* leaves, *Chenopodium ambrosioides* L. essential oil or *Nigella sativa* essential oil [10,29,32].

This was further supported by the accumulation of ethidium bromide observed in the strain overexpressing the efflux system (*S. aureus* SA1199B) (Figure 2A). Interestingly, in the presence of resveratrol, ethidium bromide accumulation was significantly increased, in line with the profile obtained for the control CCCP. This fact highlights the EPI role of resveratrol in reducing antimicrobial resistance or tolerance by increasing the intracellular accumulation of antimicrobials. In fact, a previous study showed that resveratrol increased susceptibility to chlorhexidine with increased intracellular accumulation of ethidium bromide in *Acinetobacter baumannii*, indicating that resveratrol acts as an AdeB EPI [26]. The comparative accumulation of ethidium bromide by the various strains with different levels of NorA expression further supports the effect of resveratrol on the efflux pump inhibition, with the *S. aureus* SA1199B strain, when in the presence of resveratrol, reaching fluorescence values similar to its native correspondent strain (SA1199). For the wildtype and the NorA-overexpressing strains, a significant increase in fluorescence associated with augmented ethidium bromide accumulation when in the presence of resveratrol was observed (Figure 2B). The exception was the *S. aureus* SAK1758 strain, which presented a *norA* deletion, indicating a lower effect of resveratrol and suggesting an effect on the NorA system. Despite this trend, the effect of the compound on other efflux systems cannot be excluded, as its potential effect as an EPI has also been described for other efflux pump families beyond the MFS superfamily, such the RND or PACE families [24,26,27].

For a better understanding of the in vitro results, molecular docking studies were performed. Since the NorA crystal structure is not available, a preliminary docking study of the NorA homologous structure (AlphaFold code: P0A0J7) and reserpine was needed to predict the best binding site on the pump (Figure 3) to further carry out the docking with resveratrol (Figure 4 and Table 2). Comparing with reserpine, resveratrol does not show higher affinity to NorA nor similar interactions (e.g., electrostatic or non-classic H-bond interactions) with the amino acids in reserpine’s binding site. Nonetheless, the higher number of classic H-bonds between resveratrol and the residues ASP307, ARG310, GLN255, ASN137 and PHE140 suggest a more stable interaction between this ligand and the receptor. These results are corroborated by the works of Kalia et al. (2012), since similar interactions with PHE140, ILE244, GLY248 and PHE303 residues were predicted [28].

Considering an effective clinical application, antimicrobials must be able to suppress mutations and resistance mechanisms [31]. Therefore, the impact of resveratrol on resistance emergence was also evaluated using *S. aureus* SA1199, since it does not have any known mutation in the regulatory region of NorA or in the targets of antibiotic action (DNA gyrase and topoisomerase IV) [28,31]. In fact, despite the fact that the mutations were not confirmed, it can be suggested that resveratrol has been shown to reduce the mutation frequency of norfloxacin in *S. aureus*. Such a situation has been previously described by Nøhr-Meldgaard et al. (2018), suggesting the co-administration of resveratrol and gentamicin to limit the development of resistance in *S. aureus* [21]. 

The PAE represents the suppression of bacterial growth after a short exposure of the bacteria to antimicrobial agents [28]. Resveratrol was shown to enhance the norfloxacin PAE with a significant increase found for norfloxacin at all the tested combinations. Thus, resveratrol promotes PAE potentiation, decreasing the ability of the bacterial cells to recover. Also, other natural compounds have shown similar potential, such as capsain, the major constituent of hot chili, or citral [28,33].

In this sense, we demonstrated a potential clinical relevance of combining this compound with the antibiotic, acting as a putative EPI, restricting the emergence of resistance and prolonging the PAE.

In sum, the obtained results indicate that resveratrol, in addition to its numerous beneficial properties for human health, exhibited clinically relevant antimicrobial activity and was able to modulate norfloxacin resistance or tolerance by inhibiting the NorA efflux pump, increasing the efficacy of the antibiotic against *S. aureus*. Furthermore, we demonstrated a potential clinical relevance of combining this compound with the antibiotic to restrict the emergence of resistance and prolong PAE. Thus, it can become a promising drug adjuvant to circumvent antibiotic resistance.

## 4. Materials and Methods

### 4.1. Bacterial Strains and Antimicrobial Agents

*S. aureus* SA1199B, which over-expressed NorA (norA++), its wildtype *S. aureus* SA1199, strain SAK1758, which has a NorA deletion mutant of NCTC 8325-4 (ΔnorA), and its wildtype NCTC 8325-4 were used in this study and were kindly provided by Professor Dr. Lorena Tuchscherr de Hauschopp (Jena University Hospital-Germany). Resveratrol (TCI Europe, Zwijndrecht, Belgium) was dissolved in dimethyl sulfoxide (DMSO), norfloxacin (Alfa Aesar, Kandel, Germany) was dissolved in water basified with sodium hydroxide and ethidium bromide (Fisher Scientific, Brussels, Belgium) was obtained as an aqueous solution.

### 4.2. Antibacterial Activity Analysis through Determination of Minimum Inhibitory Concentration (MIC) and Modulatory Activity of Resveratrol

The MIC of resveratrol, norfloxacin and ethidium bromide was determined using broth microdilution tests, according to the guidelines of the Clinical and Laboratory Standards Institute [34]. Briefly, serial two-fold dilutions of the compounds in Mueller Hinton Broth (MHB) were conducted in a 96-well microplate and a bacterial suspension was added to a final volume of 100 μL at a final concentration of ~5 × 10^5^ colony-forming units (CFU)/mL. The concentration range of the compounds under study was adjusted according to the susceptibility of the strains. After incubation at 37 °C for 24 h, the MIC value was assessed as the lowest concentration of each compound that inhibited the visible growth of bacteria. 

The resveratrol modulatory effect was evaluated using the MIC determination for norfloxacin and ethidium bromide in the presence of a sub-MIC level (0.25× MIC) of resveratrol, according to the method described by Diniz-Silva and collaborators (2017) [30]. All the assays were performed in duplicate in at least three independent tests, and the modal values were selected for further assays.

### 4.3. Ethidium Bromide Accumulation Assay

Ethidium bromide accumulation was analyzed as described by Espinoza et al. (2019) with some modifications [7]. Briefly, the *S. aureus* SA1199B cells were grown in MHB at 37 °C and 250 rpm until the exponential phase (OD_600nm_~0.6), then they were collected through centrifugation (10,000× *g*, 3 min) and washed twice with phosphate-buffered saline (PBS) (pH = 7.2). The cell deposit was then resuspended in PBS with 0.6% (*w*/*v*) glucose, adjusting to OD_600nm_ of 0.3. The suspension was transferred to a black 96-well microplate with a clear bottom (Greiner Bio-One, Frickenhausen, Germany) and resveratrol (final concentration of 0.25× MIC), ethidium bromide (final concentration of 2 mg/L) and/or CCCP (final concentration of 1.25 mg/L) were added. Fluorescence was then monitored at 37 °C every 3 min for a period of 1 h in a Spectramax Gemini XS spectrofluorometer (Molecular Devices LLC, San Jose, CA, USA) at excitation and emission wavelengths of 530 nm and 580 nm, respectively. For *S. aureus* strains SA1199, NCTC 8325-4 and SAK1758, the fluorescence measure was taken at 60 min. The assays were performed in triplicate, with at least three independent tests.

### 4.4. Molecular Docking Studies with Resveratrol

Docking of reserpine and resveratrol to NorA was conducted in AutoDock Tools (ADT) 1.5.6 software (The Scripps Research Institute, San Diego, CA, USA), similarly to the work performed by other authors [28,31]. The 3D predicted NorA structure was obtained from the AlphaFold Protein Structure Database (alphafold.ebi.ac.uk, accessed on 20 March 2022) under the code P0A0J7 [35,36]. The minimized energy 3D structures of reserpine and resveratrol were obtained using ChemDraw and Chem3D software at a minimum RMS gradient of 0.01 and saved in “.pdb” format. In ADT, all the hydrogen atoms were added, Gasteiger charges were computed, non-polar hydrogens were merged and AD4 type atoms were assigned to protein structure. Flexible and rigid bonds were identified in reserpine and resveratrol. The grid box used for the docking of reserpine to NorA was defined with the coordinates (x, y, z) 2.392, 0.446, (−0.965) and the size (x, y, z) 126 × 126 × 126 points with 0.375 Å of spacing covering the whole receptor structure. The grid box used for the docking of resveratrol to NorA was defined with the coordinates (x, y, z) 7.303, 1.676, (−0.068) and the size (x, y, z) 35 × 35 × 35 points with 0.375 Å of spacing.

Docking was performed using a Lamarckian genetic algorithm with a total of 20 runs, and the conformation of each compound with the lowest estimated binding energies (BE, Kcal/mol) was considered the most optimal conformation and used for analysis.

### 4.5. Impact on Frequency of Resistance

In order to evaluate the impact of resveratrol on the emergence of norfloxacin resistance in *S. aureus*, a study was performed according to the method of Singh et al. (2017) with minor modifications [31]. A bacterial suspension (*S. aureus* SA1199) containing ~10^10^ CFU/mL was prepared. Next, 0.1 mL was spread onto Mueller Hinton Agar (MHA) plates supplemented with 4×, 8× and 16× MIC of norfloxacin concentrations with or without 0.25× MIC of resveratrol (50 mg/L). The total count of viable cells was performed using successive dilutions from the cell suspension. After 48 h incubation at 37 °C, the number of colonies was counted, and the mutation frequency was calculated. Mutation frequency was determined as the proportion of colonies appearing in norfloxacin or norfloxacin combined with resveratrol to total colonies (CFU/mL) plated. At least three independent tests were carried out.

### 4.6. Post-Antibiotic Effect

For PAE evaluation, 5 mL of MHB containing different concentrations of norfloxacin (0.25×, 0.5× and 1× MIC), with and without 25 mg/L of resveratrol, was inoculated with 0.05 mL of a suspension of *S. aureus* SA1199B at 0.5 McFarland. After 2 h of incubation at 37 °C and 250 rpm, samples (0.005 mL) were collected in test tubes with 5 mL of MHB (dilution 1:1000) to effectively remove the drug and resveratrol. Thereafter, viability counts were determined on MHA plates at 0 h and every 1 h until the control tube became turbid. PAE was defined as PAE = T − C, where (T) represents the growth time of the exposed culture and (C) represents the unexposed control, where there is an increase in population growth by 1 log_10_ CFU/mL after exposure of *S. aureus* SA1199B to antimicrobial agents [28]. At least three independent assays were conducted.

## Figures and Tables

**Figure 1 antibiotics-12-01168-f001:**
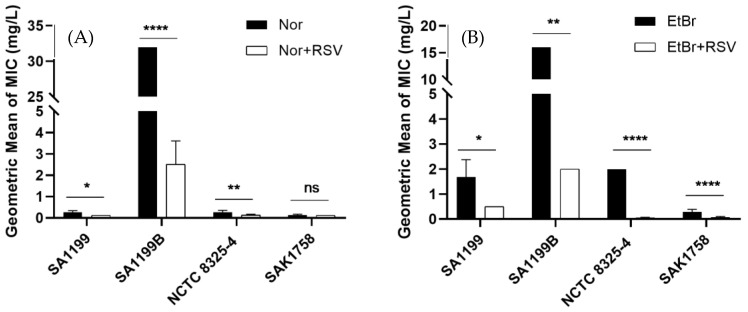
Minimum inhibitory concentration of (**A**) norfloxacin and (**B**) ethidium bromide in the presence and absence of resveratrol (0.25× MIC) against *S. aureus* with different levels of NorA expression. The values are expressed as the geometric mean and geometric standard deviation of at least three independent experiments. * *p* < 0.05; ** *p* < 0.01; **** *p* < 0.0001; ns: no significance.

**Figure 2 antibiotics-12-01168-f002:**
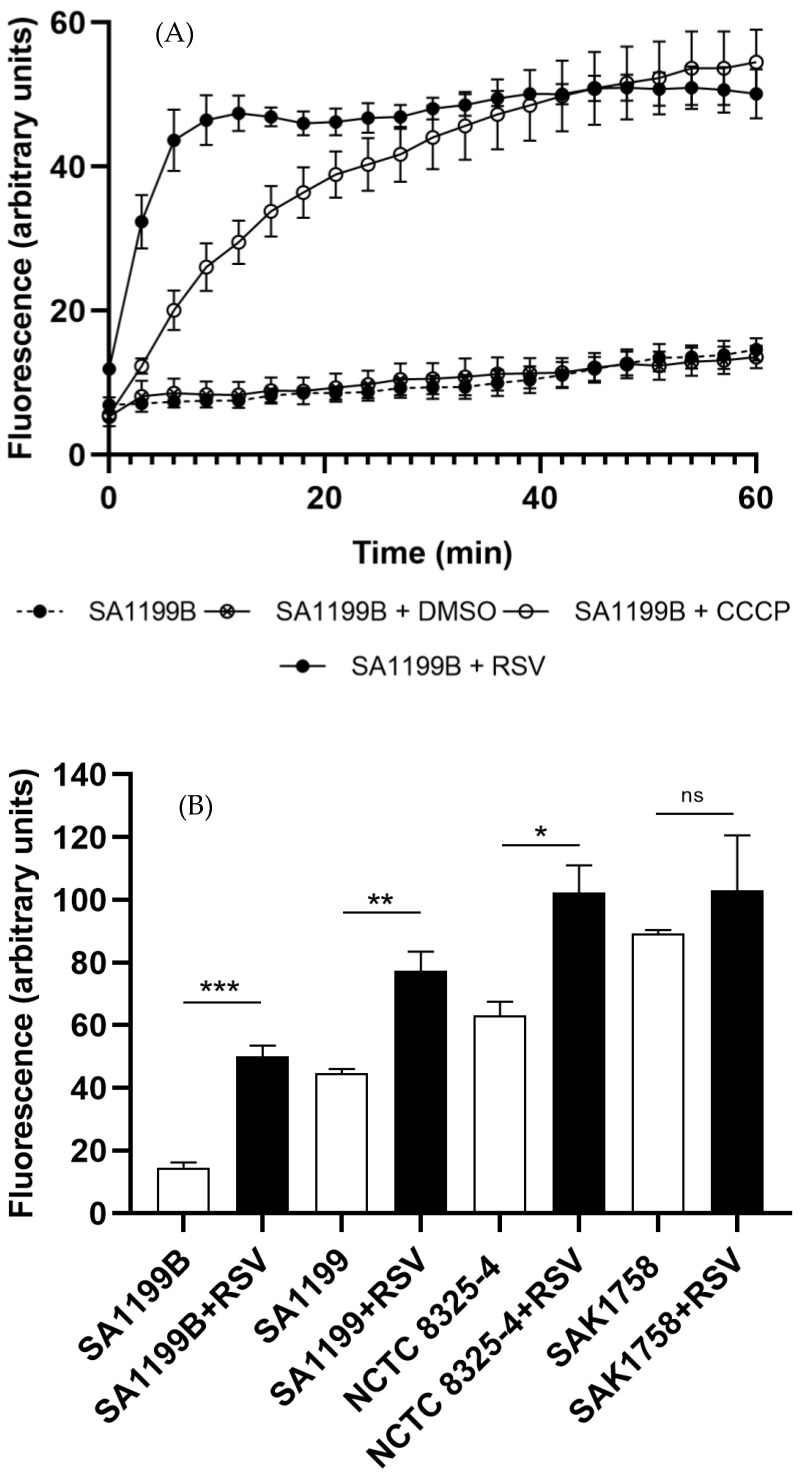
Effect of resveratrol (RSV) on intracellular accumulation of ethidium bromide (**A**) in *S. aureus* SA1199B (norA++) in the presence and absence of resveratrol (0.25× MIC). The control with CCCP (1.25 mg/L), a known EPI, and the DMSO (0.25%) solvent control are also represented. (**B**) Comparative ethidium bromide accumulation in *S. aureus* with different levels of NorA expression, SA1199B (norA++), SA1199 (wildtype), NCTC 8325-4 (wildtype) and SAK1758 (∆norA), after 60 min of accumulation in presence and absence of resveratrol (0.25× MIC). * *p* < 0.05; ** *p* < 0.01; *** *p* < 0.001; ns: no significance.

**Figure 3 antibiotics-12-01168-f003:**
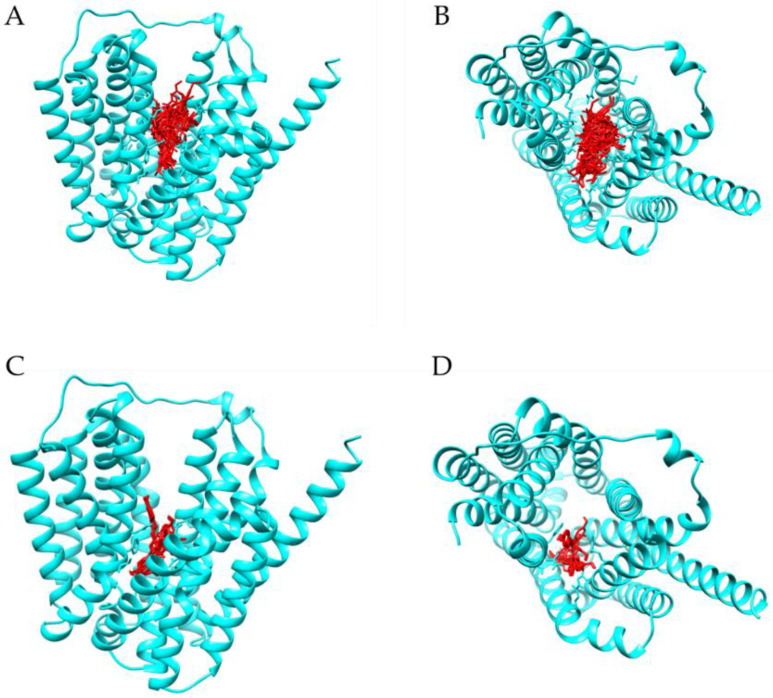
Different orientations of reserpine (red, (**A**,**B**)) and resveratrol (red, (**C**,**D**)) obtained in the docking with *Staphylococcus aureus* NorA efflux pump (blue). Using a grid covering the entire protein, it is possible to observe that all conformations are presented in the same location, suggesting a credible binding site of reserpine in the NorA pump.

**Figure 4 antibiotics-12-01168-f004:**
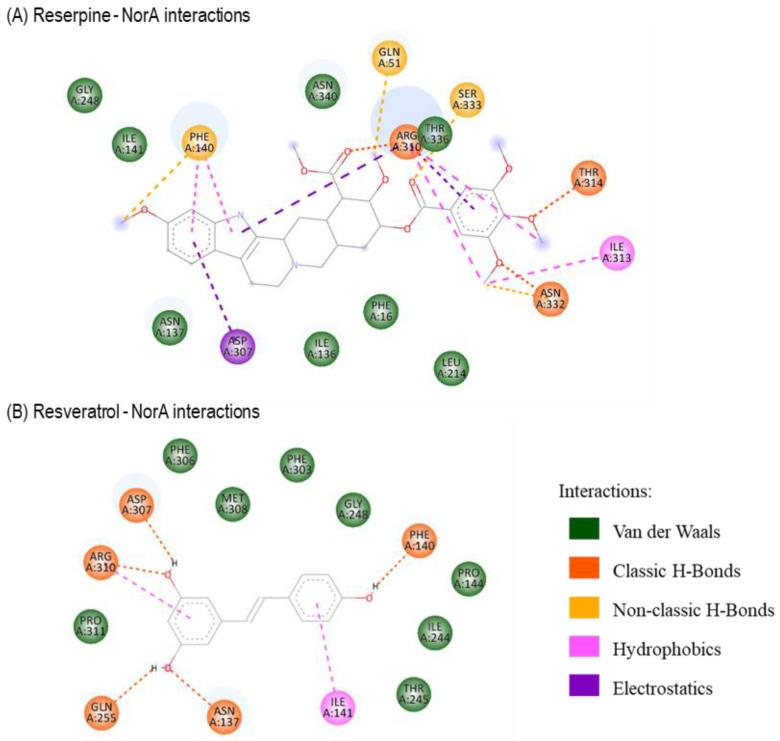
Two-dimensional perspective of the interactions between reserpine (**A**) and resveratrol (**B**) with NorA efflux pump. Each color represents the type of interaction.

**Table 1 antibiotics-12-01168-t001:** Minimum inhibitory concentrations (MIC) of resveratrol against the *S. aureus* strains.

Bacterial Strains	MIC (mg/L) of Resveratrol
SA1199 (wildtype)	200
SA1199B (*norA*++)	100
NCTC 8325-4 (wildtype)	400
SAK1758 (*ΔnorA*)	200

**Table 2 antibiotics-12-01168-t002:** In silico estimated free binding energy and interactions of reserpine and resveratrol to NorA pump.

Ligand		Reserpine	Resveratrol
Binding Energy (Kcal/mol)		−7.39	−6.19
Interactions	Van der Waals	LEU214 PHE16 ILE136 ASN137 GLY248 ILE141 ASN340 THR336	PHE306 MET308 PHE303 GLY248 PRO144 ILE244 THR245 PRO311
	Classic H-Bonds	ASN332 THR314 ARG310	ASP307 ARG310 GLN255 ASN137 PHE140
	Non-classic H-Bonds	PHE140 GLN51 SER333	----
	Hydrophobics	ILE313 ARG310 PHE140	ILE141 ARG310
	Electrostatics	ASP307 ARG310	----

**Table 3 antibiotics-12-01168-t003:** Mutation frequency of norfloxacin with resveratrol against *S. aureus* SA1199 in the absence and presence of resveratrol (0.25× MIC) for different concentrations of norfloxacin (4×, 8× and 16× MIC).

Resveratrol (mg/L)	Mutation Frequency with Norfloxacin (±SD)		
	4× MIC (1 mg/L)	8× MIC (2 mg/L)	16× MIC (4 mg/L)
0	1.87 (±0.39) × 10^−5^	5.27 (±6.07) × 10^−7^	4.03 (±1.98) × 10^−8^
50	2.16 (±1.27) × 10^−7^	1.93 (±1.33) × 10^−8^	<5.31 × 10^−10^

**Table 4 antibiotics-12-01168-t004:** PAE of norfloxacin (Nor) alone and in combination with resveratrol (RSV) against *S. aureus* SA1199B after exposure of 2 h.

Regimen	Mean PAE (h) ± SD		
	0.25× MIC Nor (8 mg/L)	0.5× MIC Nor (16 mg/L)	1× MIC Nor (32 mg/L)
**Nor**	0.04 ± 0.08	0.07 ± 0.04	0.48 ± 0.14
**Nor + RSV (25 mg/L)**	0.75 ± 0.09 ***	0.63 ± 0.06 ***	0.93 ± 0.22 *

* *p* < 0.05; *** *p* < 0.001 (*t-Student test*).

## Data Availability

Data are contained within the text.

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
