# Peer review of "Resveratrol as an Inhibitor of the NorA Efflux Pump and Resistance Modulator in Staphylococcus aureus"

_antibiotics, 2023, doi:10.3390/antibiotics12071168_

Round 1

Reviewer 1 Report

Thank you for your submission to Antibiotics. The work presents the ability evaluation and the possible mechanism of resveratrol to modulate norfloxacin resistance in S. aureus. The manuscript fit within the scope of Antibiotics. The document is well written and as required by the journal. I think that the manuscript may be considered for publication on Antibiotics after the major revision. The comments are as follows:

1.      The introduction muse be improved by providing sufficient background and include all relevant references.

2.      Please add the adequate evidence or results to support the present work.

Well written.

Author Response

We appreciated the careful review and greatly acknowledge the comments provided by the reviewer. The reviewer suggested some changes in order to improve the manuscript. The changes made are detailed in the following items and highlighted in yellow and green in the manuscript.

General comments: Thank you for your submission to Antibiotics. The work presents the ability evaluation and the possible mechanism of resveratrol to modulate norfloxacin resistance in S. aureus. The manuscript fit within the scope of Antibiotics. The document is well written and as required by the journal. I think that the manuscript may be considered for publication on Antibiotics after the major revision. The comments are as follows:

Comment 1.      The introduction muse be improved by providing sufficient background and include all relevant references.

Response: As suggested, the introduction section was vastly reviewed and improved.

Comment 2.      Please add the adequate evidence or results to support the present work.

Response: Accordingly with the reviewer’ suggestion, we added more controls to the paper to better support the work (marked in green along the paper), and also added more discussion regarding our results.

Reviewer 2 Report

The Research is relevant and well-designed. However, the Research on efflux pump inhibitors is complex, so more precise controls in the experiments and deeper literature analysis linking Your results with the results obtained by other researchers are required to avoid misinterpretations—more comments in the attached file.

Author Response

We appreciated the careful review and greatly acknowledge the comments provided by the reviewer. The reviewer suggested some changes in order to improve the manuscript. The changes made are detailed in the following items and highlighted in blue in the manuscript.

General comments: The Research is relevant and well-designed. However, the Research on efflux pump inhibitors is complex, so more precise controls in the experiments and deeper literature analysis linking Your results with the results obtained by other researchers are required to avoid misinterpretations—more comments in the attached file.

Abstract.

Comment: The abstract must be 200 words or less. It should be more specific without many connective words. Make it shorter.

Response: The reviewer’s comments was followed by reducing the number of words and connective words.

Introduction

Comment 1. 49-51: Staphylococcus aureus is gram-positive bacteria. The examples where resveratrol could be a potential inhibitor of efflux pumps are from the results with gram-negative bacteria. It would be best to give examples of gram-positive bacteria, including the family of efflux pumps because the information about the efficiency of an inhibitor to the exact family/type of efflux pump is crucial.

Response: Although some research works demonstrate the ability of resveratrol to inhibit efflux pumps in some gram-negative bacteria, the literature lacks data regarding the effect on gram-positive bacteria. One of the works describing the EPI activity on gram-negative bacteria found a diminution of expression on a gene coding to a MFS family efflux pump, but to our knowledge is the only showing some role of this compound in systems from this family. We improved the information presented in the introduction regarding this subject.

Results

Comment 1. Figure 1. Indicate the concentration of compounds (DMSO, CCCP) in the figure description below.

Response: As suggested by the reviewer the concentrations used were added in the figure caption.

Comment 2. Figure 1. Y curve: Fluorescence, RFU. The values of fluorescence intensity are pretty low. Did you use arbitrary units (a.u)?

Response: Yes, we used arbitrary units, the modification was done in yy axis of the graphics.

Comment 3. Figure 1. What is a control curve of death cells? What amount of ethidium can the dead cells accumulate? A control curve is required. Because the best inhibitor is that it does not affect the viability of cells at high amounts but which can inhibit the efflux, you should prove this.

Response: No death cells were used in this assay. It was previously described that the resveratrol has a bacteriostatic effect on S. aureus, thus the assay was performed considering that with 1/4xMIC of resveratrol during one hour the cells would not be affected. This is further supported by the work of other authors who previously described that resveratrol-induced membrane damage is not detected for Staphylococcus aureus (https://doi.org/10.1016/j.ijantimicag.2018.06.005).

Regarding controls, and to support better our claims, we added a comparative analysis of accumulation of ethidium bromide for strains expressing different levels of norA (Figure 1B).

Discussion

Comment 1. More other researchers' publications need to be included in substantiating their data. You need more than one source to base your claim.

Response: The discussion was improved and more references added, however for some statements the literature does not presents a lot of data.

Reviewer 3 Report

In this study, authors studied the effect of resveratrol on S. aureus antimicrobial susceptibility, especially towards norfloxacin. The effect of resveratrol was prominent in a strain which expressed more NorA efflux pumps. This study requires more control to prove authors claim. Authors need to give more explanation and analysis of their data in the result section. If authors made different sub-section of their result, then it would be easier to follow. I have the following suggestions for authors:

1) Authors used the word "norfloxacin resistance" in their manuscript. Antimicrobial resistance is usually related to genetic change that cause antibiotic to become ineffective against the tested bacterial species. In this study, it was observed that co-treatment with resveratrol led to a decrease in tolerance for norfloxacin, or the survival of bacteria decreased upon treatment with norfloxacin. Please explain why the term resistance was used over tolerance or survival. 

2) Lines 55–63: Authors should provide the figure of raw data, which includes SD and statistical significance.

Authors did not mention the result of control SAK1758 (ΔnorA) in the text, please explain it. If authors complement the SAK1758 strains, then their results will be enhanced.

3) Lines 66–69: Please explain how CCCP could act as a positive control in the text.

Authors should include WT and SAK1758 in Figure 2 and compare the results.

4) Lines 86-89, figure 2: Put docking result for NorA and resveratrol, just like NorA and reserpine.

Table 2: If resveratrol's binding to NorA is similar to reserpine, then why there is no similarity in interacting amino acids?

5) Lines 100–105, Table 3, mutation frequency:

The MIC was calculated by using 5×105 CFU/mL. In the mutation frequency assay, the authors used 1010 CFU/mL. It may be possible that after treatment with antibiotic, the colonies obtained were just surviving cells because the starting CFU were higher, and those may not be the mutated bacteria.

To confirm that the surviving colonies were mutated, those colonies need to be reinoculated in the presence of antibiotics and checked for regrowth.

In the discussion (Lines 155–166), as mutations in the surviving colonies were not confirmed, please modify the discussion or provide additional supporting data.

Keep the other two strains as a control.

Line 119, Table 4: Why statistical significance was shown only at 0.5 x MIC concentration.  Author should give raw data for the same in manuscripts.

In discussion (lines 167–175), authors should explain why they don’t see the PAE effect at higher MIC.

Line 152: There is no figure 3 in the text.

Author Response

We appreciated the careful review and greatly acknowledge the comments provided by the reviewer. The reviewer suggested some changes in order to improve the manuscript. The changes made are detailed in the following items and highlighted in green in the manuscript.

General comments: In this study, authors studied the effect of resveratrol on S. aureus antimicrobial susceptibility, especially towards norfloxacin. The effect of resveratrol was prominent in a strain which expressed more NorA efflux pumps. This study requires more control to prove authors claim. Authors need to give more explanation and analysis of their data in the result section. If authors made different sub-section of their result, then it would be easier to follow. I have the following suggestions for authors:

Comment 1) Authors used the word "norfloxacin resistance" in their manuscript. Antimicrobial resistance is usually related to genetic change that cause antibiotic to become ineffective against the tested bacterial species. In this study, it was observed that co-treatment with resveratrol led to a decrease in tolerance for norfloxacin, or the survival of bacteria decreased upon treatment with norfloxacin. Please explain why the term resistance was used over tolerance or survival. 

 Response: The term resistance was used since in the case of SA1199B the use of resveratrol restored the susceptibility of the strain to norfloxacin. Nonetheless, the manuscript was reviewed and the term resistance changed for tolerance when the situation was appropriated.

Comment 2) Lines 55–63: Authors should provide the figure of raw data, which includes SD and statistical significance.

Response: Regarding the MIC determination in absence and presence of resveratrol, the modal values were used. This information was added to the Materials and Methods section.

Comment 3) Authors did not mention the result of control SAK1758 (ΔnorA) in the text, please explain it. If authors complement the SAK1758 strains, then their results will be enhanced.

 Response: The information related to the SAK1758 strain has been added to the text. Despite not having the complemented strain, its native strain showed a small decrease in the MIC value of norfloxacin when in presence of resveratrol, supporting the effect on NorA, such as observed with SA1199 and SA1199B.

Comment 4) Lines 66–69: Please explain how CCCP could act as a positive control in the text.

Response:  We considered CCCP as a positive control, since it is a known EPI, however, we understand that this may be misleading and changed the expression.

Comment 5) Authors should include WT and SAK1758 in Figure 2 and compare the results.

Response: Such as requested by the reviewer the data from the strains was included in the Figure 1, by adding a new graphic with the three strains with and without resveratrol. Some discussion on this was also added to the manuscript.

Comment 6) Lines 86-89, figure 2: Put docking result for NorA and resveratrol, just like NorA and reserpine.

Response: As suggested by the reviewer, the different orientations of resveratrol obtained in the docking with Staphylococcus aureus NorA efflux pump were added to Figure 2.

Comment 7) Table 2: If resveratrol's binding to NorA is similar to reserpine, then why there is no similarity in interacting amino acids?

Response: Comparing with reserpine, resveratrol does not show neither higher affinity to NorA nor any similar interactions (e.g. electrostatic or non-classic H-Bonds interactions) with the amino acids in reserpine's binding site. This information was added to the manuscript in order to clarify.

Comment 8) Lines 100–105, Table 3, mutation frequency:

The MIC was calculated by using 5×105 CFU/mL. In the mutation frequency assay, the authors used 1010 CFU/mL. It may be possible that after treatment with antibiotic, the colonies obtained were just surviving cells because the starting CFU were higher, and those may not be the mutated bacteria. To confirm that the surviving colonies were mutated, those colonies need to be reinoculated in the presence of antibiotics and checked for regrowth.

In the discussion (Lines 155–166), as mutations in the surviving colonies were not confirmed, please modify the discussion or provide additional supporting data.

Keep the other two strains as a control.

Response: The cells concentration used in this assay was adjusted and augmented relatively to the amount of cells used in the MIC determination assay, since, such as it can be observed by the frequency of mutation values, if using lower cell concentrations we would have mostly below limit results. Despite not confirming the mutations on the manuscript, this assay is vastly used for bacteria to assess the mutation frequency with different purposes. The information that we didn’t confirm the mutations was added to the manuscript.

The strain SA1199 was used for this assay, since it does not have any known mutation in the regulatory region of NorA and in the targets of antibiotic action (DNA gyrase and topoisomerase IV), and it is a strain that has been described for these procedures.

Comment 9) Line 119, Table 4: Why statistical significance was shown only at 0.5 x MIC concentration.  Author should give raw data for the same in manuscripts. and Comment 10) In discussion (lines 167–175), authors should explain why they don’t see the PAE effect at higher MIC.

Response: We would like to thank the careful revision that was made by the reviewer, in fact by mistaken the values in the table were concerning to the average time that each culture exposed to the compounds took to increase by  1 log10 cfu/mL immediately after compound removal, instead of PAE that corresponds to the difference in the time for growth in the exposed culture (T) and the corresponding unexposed control (C) to increase by 1 log10 cfu/mL after compound removal. This was corrected, and now it can be seen a statistical difference with and without resveratrol for all the tested concentrations.

Comment 11) Line 152: There is no figure 3 in the text.

Response: We are sorry for the error. In fact, the figure was attached to in the submission, but not added to the manuscript. This was corrected and now you can see the figure integrated in the text.

Round 2

Reviewer 1 Report

Dear Editor,

Please accept in present form.

Author Response

We appreciated the careful review and greatly acknowledge the comments provided by the reviewers.

Reviewer 2 Report

The manuscript improved and can be published. 

Author Response

(The authors gave the same response as above.)

Reviewer 3 Report

Authors should provide the representative raw data for Table 1 results in figure format with statistical significance.

If appropriate with the text, then please use either antimicrobial tolerance or resistance.

Author Response

We appreciated the careful review and greatly acknowledge the comments provided by the reviewer. The reviewer suggested some changes in order to improve the manuscript. The changes made are detailed in the following items and highlighted in green in the manuscript.

Comment 1. Authors should provide the representative raw data for Table 1 results in figure format with statistical significance.

Response: As requested by the reviewer, a figure with the geometric mean of the MIC for norfloxacin and ethidium bromide in presence and absence of resveratrol was added to the manuscript and the text revised accordingly.

Comment 2. If appropriate with the text, then please use either antimicrobial tolerance or resistance.

Response: As suggested by the reviewer the term antimicrobial tolerance or resistance was adjusted along the text.